# Exploiting Benford's Law for Weight Regularization of Deep Neural Networks

**Julius Ott**                                                      *juliusg.ott@tum.de*
*Technical University of Munich*
*Infineon Technologies AG*

**Huawei Sun**                                                     *huawei.sun@tum.de*
*Technical University of Munich*
*Infineon Technologies AG*

**Enrico Rinaldi**                                              *erinaldi.work@gmail.com*
*Riken iTHEMs*

**Gianfranco Mauro**                                       *gianfranco.mauro@infineon.com*
*Infineon Technologies AG*

**Lorenzo Servadei**                                         *lorenzo.servadei@tum.de*
*Technical University of Munich*

**Robert Wille**                                                *robert.wille@tum.de*
*Technical University of Munich*

**Reviewed on OpenReview:** *https: // openreview. net/ forum? id= TnT59yz7lc*

## Abstract

Stochastic learning of Deep Neural Network (DNN) parameters is highly sensitive to training strategy, hyperparameters, and available training data. Many state-of-the-art solutions use weight regularization to adjust parameter distributions, prevent overfitting, and support generalization of DNNs. None of the existing regularization techniques have ever exploited a typical distribution of numerical datasets with respect to the first non-zero (or significant) digit, called Benford's Law (BL). In this paper, we show that the deviation of the significant digit distribution of the DNN weights from BL is closely related to the generalization of the DNN. In particular, when the DNN is presented with limited training data. To take advantage of this finding, we use BL to target the weight regularization of DNNs. Extensive experiments are performed on image, tabular, and speech data, considering convolutional (CNN) and Transformer-based neural network architectures with varying numbers of parameters. We show that the performance of DNNs is improved by minimizing the distance between the significant digit distributions of the DNN weights and the BL distribution along with L2 regularization. The improvements depend on the network architecture and how it deals with limited data. However, the proposed penalty term improves consistently and some CNN-based architectures gain up to 15% test accuracy over the default training scheme with L2 regularization on subsets of CIFAR 100.

## 1    Introduction

The advent of *Deep Neural Networks* (DNNs) has revolutionized several domains by exploiting their adaptability and robust learning capabilities without requiring full model interpretability. Although the number of large datasets in DNN research is constantly increasing, many industrial applications must deal with smaller, manually collected datasets. In addition, the choice of hyperparameters significantly affects the

training stability and overall performance.

A notable area of research is weight regularization. Over the years, numerous studies have shown how explicit weight regularization described in Van Laarhoven (2017), such as L2 or L1 regularization, can improve learning by imposing penalties to encourage smaller or sparser weight distributions. Implicit regularization techniques, such as *Mixup* by Zhang et al. (2018) or *Cutmix* by Yun et al. (2019), modify the input data to achieve similar effects. These strategies highlight the importance of weight regularization in improving the learning capabilities of DNNs. However, they are scale dependent and need to be fine-tuned for individual domains.

This paper explores a remarkable aspect of DNNs - the distribution of the first significant digits of their weights, a phenomenon closely related to Benford's Law (BL). Discovered independently by Newcomb (1881) and Benford (1938), BL describes a counterintuitive but common pattern in numerical data across several domains: Lower significant digits occur more often than higher ones.

This scale-invariant pattern described by BL has been observed in various domains, such as physical constants by Shao & Ma (2010) and stock prices by Ley (1996). In machine learning, the properties of BL have been exploited to detect anomalies in input data by O'Mahony et al. (2023) and synthetically generated images by Bonettini et al. (2021), underlining its relevance. Their motivation is derived from the observation that natural datasets tend to exhibit the pattern of BL, while synthetic data does not demonstrate this pattern. Sahu et al. (2021) show that the link between BL and DNN weights is based on their mutual relationship to thermodynamics. In particular, the probability of energy states in closed systems is such that smaller energy states are more likely to occur than larger ones. This pattern is consistent with that observed for BL.

Despite the highlighted correlations between DNNs and the Benford distribution, research has only focused on observing whether the collected significant digits of trained weights follow BL. The authors of Sahu et al. (2021) propose to use the correlation between significant digits of the weights and BL to estimate the generalization error of DNNs. A high correlation is then used as a stopping criterion for *Early Stopping*, a mechanism that stops training when performance on a holdout validation dataset does not increase for a predefined number of consecutive epochs. This feature of BL is closely related to its relevance for bias detection in numerical data, such as synthetic images. In particular, data, especially with limited quantity, is always biased because it represents a small snapshot of the real world. In the same way, this bias is reflected in the weights of DNNs. Consequently, a DNN trained with limited training data is expected to deviate from the ideal case of bias-free learning. One potential solution to enhance the performance of the DNN is to bias the weights towards the ideal case of bias-free learning by incorporating a penalty term into the loss function.

To date, no research has been published on the use of BL as a regularization technique. To analyze the effects of BL on DNNs weights regularization, this paper:

1. demonstrates the relation between ML datasets and BL

2. introduces a way to approximate BL via gradient-based optimization,

3. analyzes the effect of Benford regularization in training DNNs with reduced datasets compared to their standard L2 regularized training scheme.

Benford regularization has particular relevance for applications where a complex dataset is insufficient for a DNN or a DNN has limited capacity for a large dataset. To this end, our experiments utilize subsets of common image (CIFAR10/100) and audio datasets to emphasize the bias in the dataset. In this manner, we are able to draw comparisons between CNN-based architectures, such as DenseNet and ResNext, and Transformer architectures, including tiny ViT and tiny Swin Transformer, with respect to their performance on random subsets of the data. The findings reveal consistent enhancements due to the Benford regularization when employed in conjunction with L2 regularization, as opposed to L2 regularization alone. In particular, CNN-based architectures benefit from additional Benford regularization in subsets of the datasets with improvements of up to 15%. Similarly, Transformer models exhibit enhancements, achieving up to 3% improvement through the integration of Benford and L2 regularization on the entire dataset.

Furthermore, experiments were conducted with small and hardware-optimized MobileNetV3 models on the Imagenet1K dataset. It was observed that the larger-capacity model demonstrated a greater improvement in performance when utilizing Benford regularization, in comparison to the smaller model. The experimental

results indicated an interaction between enhanced performance, model capacity, and data availability when employing Benford regularization.

The remainder of the manuscript is structured as follows. First, an investigation is conducted into related research on weight regularization and BL in connection to DNNs. Subsequently, an outline is presented of the relation between BL and physics, with an illustration of a close connection to DNNs. The proposed method introduces a differentiable approximation of the Benford distribution and demonstrates how BL can be approximated via gradient descent. The following experiments are performed on the public image datasets CIFAR10/100 (Krizhevsky, 2009), and Imagenet1K (Russakovsky et al., 2015). These experiments evaluate the performance of well-known DNNs trained from scratch with different dataset sizes. To complete the experiments, the regularizer is evaluated on different data domains, such as speech and tabular data. Following a thorough discussion of the experiments and their results, the limitations of the presented work are assessed. Finally, the paper concludes with an overview of its findings.

## 2    Considered problem and related work

Research on weight regularization focuses on specific favorable weight distributions. The goal of these approaches is to make DNN training stable while ensuring robustness against overfitting and noise in the data. This section discusses state-of-the-art research that employs regularization and data augmentation for DNNs, and related work that features BL with respect to DNNs.

**Regularization and data augmentation**  are of practical importance when training DNNs to avoid overfitting and foster numerical stability. Weight regularization, like L2 or L1, aims to reduce the norm of the weights by adding a penalty to the loss function. L2 regularization is mainly implemented as weight decay as described in Loshchilov & Hutter (2019). Alongside L2 regularization, the focus of seminal work is on implicit regularization, such as *data augmentation* methods reviewed by Shorten & Khoshgoftaar (2019) like *Mixup* by Zhang et al. (2018) or *Cutmix* by Yun et al. (2019). Such techniques do not directly constrain the weights, but they affect their distribution. *Data augmentation* distorts the input data to make the network robust against different conditions, such as lighting conditions for camera images, thus improving generalization. To achieve this, *Mixup* fuses two training samples and their respective label via a convex combination to learn smooth decision functions. *Cutmix* extends the approach and cuts out a patch from an input image and replaces the patch with information from a different training sample. The targets are adjusted proportionally to number of switched pixels. An alternative regularization method, *SAM* by Foret et al. (2019), reduces the sharpness of the loss landscape via second order gradient smoothing. Generalization is achieved by finding parameters that lie in a neighborhood with a low loss instead of suboptimal sharp minima. At last, *Early Stopping* is one of the most common methods in DNN research. It describes a mechanism to stop the training when a monitored metric, like validation error, stops improving (Yao et al., 2007). In particular, *Early Stopping* does not change the weights by adding a penalty or changing the input data, so it is of relevance for training DNNs in any data domain and is commonly used. In the work of Sahu et al. (2021), the deviation to BL has been used as metric when to stop the training. This metric makes a validation dataset obsolete, which in turn can be used as additional training data.

**The distribution of significant digits**  has not yet been used to regularize the weights of DNNs via gradient-based optimization. To date, BL has been used to analyze the characteristics and the liability of the data. In Bonettini et al. (2021), the authors use the significant digits of the cosine transformation coefficients of an image to detect whether it was generated by a *Generative Adversarial Network* (GAN). Similarly, BL has been used in O'Mahony et al. (2023) to discriminate between natural and corrupt data, leveraging it as a filter to detect out-of-distribution data or anomalous data points. These methods monitor whether the collected data or generated images follow BL and are motivated by the observation that natural datasets are known to obey BL. As demonstrated by Sahu et al. (2021), the distribution of significant digits is a predictor of the generalization of DNNs to the validation data and can function as an *Early Stopping* criterion. Notably, the incorporation of BL into the optimization process has been overlooked, consequently missing significant aspects of the Benford distribution.

## 3 Motivation

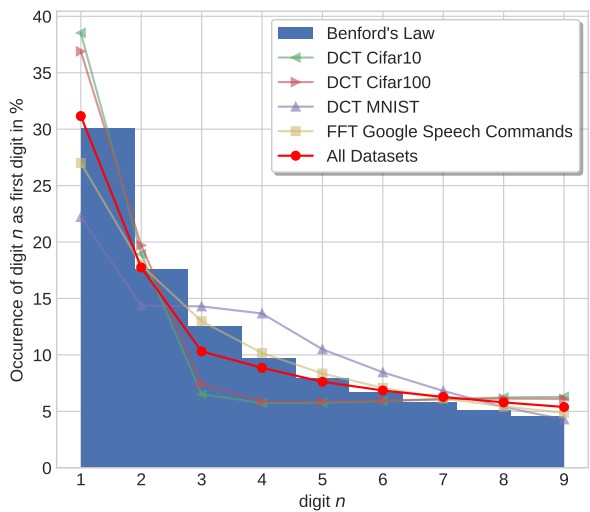

Figure 1: Comparison of significant digit distribution of DCT/FFT coefficients in public image and speech datasets, the union of datasets, and Benford's Law.

Regularization and data augmentation are two practical techniques with the potential to enhance the training and generalization of DNNs. However, these techniques must be customized to the specific data being utilized, which can lead to a non-universal approach.

In various domains, researchers have observed that the significant digits of datasets often follow a peculiar pattern, as shown in Figure 1, known as BL. Despite its widespread occurrence, the underlying reason behind BL remains a mystery (Wang & Ma, 2023). Interestingly, random samples from randomly selected distributions have been shown to converge to BL (Hill, 1995). This finding is also supported by the original work of Benford, who noted that while some datasets deviate from BL, their union closely aligns with it (Benford, 1938), underlining that unbiased data tend to adhere to BL. To further explore this phenomenon, we analyzed the frequency spectrum of several image datasets commonly used for DNN training. The coefficients of the Discrete Cosine Transformation (DCT) and Fast Fourier Transformation (FFT) are known to follow BL, as evidenced by previous studies, see Benford (2021). As shown in Figure 1, not all frequencies of these datasets follow BL. However, the union of all datasets exhibits a strong correlation, providing further evidence that data from various unbiased distributions obey BL. Mathematically, the law is fulfilled when the frequency $P(d)$ of any significant digit $d$ is given as

$$P(d) = \log_b\left(1 + \frac{1}{d}\right), \tag{1}$$

where $b$ is the base of the number system. Alternatively, the authors in Berger & Hill (2015) demonstrate that a sequence satisfies BL if and only if "the fractional parts of its decimal logarithm are uniformly distributed between zero and one", as shown in Eq. 2

$$P(d) = \log_{10}\left(1 + \frac{1}{d}\right) \quad iff \quad \log_{10}(X) \bmod 1 \sim \mathcal{U}(0, 1), \tag{2}$$

where $X$ represents arbitrary numerical data. The described law is of particular interest because it is scale- and base-invariant, as shown by Berger & Hill (2021). To measure the deviation from BL, we compute the Kullback-Leibler (KL) divergence by (Kullback & Leibler, 1951) between the significant distribution of the weights and BL (BL KL), formally defined in Eq. 3

$$KL(Q \parallel BL) = \sum_{d=1}^{9} Q(d) \log\left(\frac{Q(d)}{P(d)}\right), \tag{3}$$

where $Q(d)$ is the observed frequency of the $d$-th significant digit in the DNN weights. In the experimental section, we show that networks trained with more data have weights closer to BL.

These observations are associated to theoretical results from Iafrate et al. (2015) on the partitioning of numbers, a subject within the realm of number theory, and its relation to BL.

**A partitioning process** describes how a fixed quantity is distributed. To illustrate, partitioning describes the energy states of particles in closed systems, see Iafrate et al. (2015) and is employed in the field of combinatorics to describe the partitioning of an integer into positive smaller integers. Common DNN structures,

such as CNNs, employ fixed filters that partition the input. Transformers, on the other hand, are employed to extract information from partitioned input data. Formally, we define a quantity $X$ that is divided into smaller parts:

$$X = \sum_j n_j x_j, \tag{4}$$

with parts $n_j$ of size $x_j$. The goal is to find the distribution $p(n)$ that explains how often we observe a part of size $x$ in the subset that adds up to $X$. Intuitively, it is obvious that smaller numbers occur more often, which can be easily verified by inspecting the subsets that add up to ten: $[\{9,1\}, \{8,2\}, \{8,1,1\}, ..., \{1,...,1\}]$. An illustrative example from Kafri (2009) describes the system of Eq. 4 with a number $X$, represented as "boxes" with $N$ non-interacting balls. The question is how many boxes have exactly $n$ balls, which in turn gives information about the probability of integers in a number. The combination of different boxes then forms a number. As the probability of a ball being in a box is the same for all boxes, the distribution of balls obeys BL. This result is further generalized by Iafrate et al. (2015), showing that the distribution of parts approximately follows a simple power law $p(n) \sim 1/x$. The significant digit distribution $P_d$ of the inverse power distribution obeys BL, as shown in Eq. 5

$$P_d = \frac{\int_{d10^p}^{(d+1)10^p} dx/x}{\int_{10^p}^{10^{p+1}} dx/x} = \frac{\ln(1+1/d)}{\ln(10)} = \log_{10}\left(1 + \frac{1}{d}\right). \tag{5}$$

The approaches of Iafrate et al. (2015) and Kafri (2009) assume equal probabilities for each ball to fall in a box or the parts of each size. This assumption is based on the maximum entropy principle by Jaynes (1957). Whenever no further information about a probability distribution is given, the maximum entropy principle is the most unbiased assumption. Finding probability distributions that maximize the entropy is used in machine learning research for regularization in Haarnoja et al. (2018) and Chiang et al. (2005). Like partitioning, DNNs for classification adopt a top-down strategy that involves the division of the input into a smaller feature space by partitioning it with the trained weights. In DNNs, the number of features represents the parts, and the weights represent their respective sizes. With sufficient training data, we expect an unbiased partitioning of the input space into the feature space, resulting in weights that are approximately Benford distributed. However, in the absence of training data, the parts are biased towards fewer features with larger weights, leading to a deviation from BL. This provides an explanation for the data obtained from Figure 2, where the lack of information in the training data leads to less diversity in the feature space and, thus, to an increase in the distance to BL.

This motivates to find a way to exploit BL for weight regularization in DNNs. Specifically, our goal is to constrain the significant digit distribution of the weights closer to BL in order to improve the test error independently of the task, focusing on smaller subsets of datasets. To the best of our knowledge, this is the first method that incorporates BL into the optimization process of neural networks via gradient-based optimization.

## 4 Approach

This section presents a framework for incorporating BL into neural network regularization. To this end, we propose a differentiable approximation of BL that can be used to optimize the significant digit distribution of the DNN weights. The significant digits of the weights are updated via gradient descent to close the distance to BL. To further improve the proposed approach, we utilize derived error functions on BL for exponential functions, thereby facilitating the relaxation of the optimization problem.

The approximation of BL is more tractable with the right-hand side of Eq. 2 as it depends on the input data $x$ and not on the significant digits. The modulo 1 operator is equivalent to the fractional part of a number. The fractional part of any positive number $x$ is defined as

$$frac(x) = x - \lfloor x \rfloor, \tag{6}$$

where $\lfloor \cdot \rfloor$ is the floor function and $\lfloor x \rfloor$ denotes the next lower integer value of $x$. This function has discontinuities at zero and one, but the gradient can be numerically approximated in those regions. To approximate the uniform distribution of the fractional part with gradient optimization, we utilize quantile regression

described in Koenker & Hallock (2001). Quantile regression has been used lately in Reinforcement Learning (RL) to learn arbitrary distributions (Dabney et al., 2018). Specifically, we compare the quantiles of the weights to the quantiles of a uniform distribution $\mathcal{U}(0, 1)$. In terms of the cumulative distribution function (c.d.f.), the quantile $p$ defines the probability that a random variable $X$ evaluates smaller or equal to a threshold $x$, as shown in Eq. 7

$$F_X(x) = Pr(X \leq x) = p. \tag{7}$$

The quantile function $Q(p)$ is defined as the inverse c.d.f, presented in Eq. 8.

$$Q(p) = F_X^{-1}(p) = \begin{cases} x_{(k)}, & k = n^{(p)} \text{ if } n^{(p)} \text{ is an integer} \\ \frac{1}{2}(x_{(k)} + x_{(k+1)}), & k = \lfloor n^{(p)} \rfloor \text{ if } n^{(p)} \text{ is not an integer} \end{cases} \tag{8}$$

Here, $x_{(1)}, x_{(2)}, \ldots, x_{(n)}$ denote the ordered weights in ascending order and $n$ is their total number. The $\lfloor \cdot \rfloor$ denotes the floor function. The expression $n^{(p)}$ is the index of the $p$-th quantile, and the value of $k$ is the integer part of $n^{(p)}$. If $n^{(p)}$ is not an integer, the $p$-th quantile is defined as the average of the $k$-th and $(k+1)$-th ordered observations. The $p$-th quantile of the standard uniform distribution $\mathcal{U}(0, 1)$ is already given by $p$. To measure the deviation from the standard uniform distribution, we compare the quantiles of the weight distribution to the quantile value, as shown in Eq. 9

$$L_{BL}(K, \theta) = \frac{1}{K} \sum_{k=1}^{K} \left( \hat{Q}(k, \theta) - k \right)^2, \tag{9}$$

where $K$ is the number of quantiles and $\hat{Q}(k, \theta)$ denotes the $k$-th quantile of the weights $\theta$. For an accurate approximation of the standard uniform distribution, the number of quantiles used for the regression is essential. The best estimation can be achieved when the number of quantiles is equal to the number of weights in the DNN. Since, quantiles are computed based on the Quickselect algorithm by Hoare (1961), which has an average complexity of $\mathcal{O}(n)$, computing them for becomes infeasible for DNNs with millions of parameters. Thus, whenever needed, the quantile regression is computed for each layer sequentially. This loss formulation is closely related to isotonic regression for DNN calibration in Niculescu-Mizil & Caruana (2005).

In Algorithm 1, we illustrate a PyTorch-style pseudocode of the quantile regression steps. Finally, the quantile regression loss is added to the objective function and consequently not limited to classification tasks.

Nevertheless, to enhance the stability of the proposed loss, we integrate the error function from Engel & Leuenberger (2003). The authors show that exponential functions of the form $\lambda e^{-\lambda x}$ obey BL within error bounds independent of $\lambda$ and $x$. The error on BL for the exponential function is defined as:

$$Er(f) = \sum_{-\infty}^{+\infty} e^{-\lambda d 10^n} (1 - e^{-\lambda 10^n}) - \log_{10}(1 + \frac{1}{d}). \tag{10}$$

According to the calculation in Engel & Leuenberger (2003), the error is bound within $Er(f) \leq 0.03$. This result can be incorporated into the proposed Benford regularizer given that the unnormalized probability density of a classification network is defined as:

$$p(x, \theta) = e^{(f(x, \theta)/\tau)}, \tag{11}$$

where $f(x, \theta)$ defines the neural network output and $\tau$ is the temperature. Therefore, losses within the error bound are neglected.

## 5 Experiments

In this section, an evaluation of the Benford regularization is conducted on various datasets and model architectures. The baseline consists of networks trained with L2 regularization, as this is an integral component of their original training scheme. The Benford regularizer is then evaluated on top of the L2 regularization,

**Algorithm 1:** PyTorch-style pseudocode of computing the quantile loss for Benford regularization

```
# W - flattened model weights of shape Nx1
def quantile_loss_BL(W):
    # Number of quantiles is equal to the number of weights N
    n_quantiles = W.shape[0]
    # Compute fractional part
    W = remainder(log10(W),1)
    # Define quantile steps between 0 and 1
    quantile_steps = linspace(start=0, end=1, steps=n_quantiles)
    # Compute quantiles for the weights
    W_quantiles = quantile(W, quantile_steps)
    # Uniform quantiles are the quantile steps
    uniform_quantiles = quantile_steps
    # Compare the quantiles with Mean Squared Error
    bl_loss = mse_loss(W_quantiles, uniform_quantiles)
return bl_loss
```

which contributes to numerical stability.

To this end, we train all networks from scratch on subsets of publicly available image datasets such as MNIST (LeCun, 1998), CIFAR 10/100 (Krizhevsky, 2009) and ImageNet1K (Russakovsky et al., 2015). In order to show invariance to different model architectures, we evaluate the Benford regularizer on CNN and Transformer models. The Transformer models used throughout the experiments have 3 times more parameters than the CNN-based networks, rendering them predisposed to overfitting on limited datasets. At the same time, they have a larger capacity which is relevant when the experiments use the entire dataset. In addition to images, the experiments evaluate the performance of the M5 model from Dai et al. (2017) on the Google Speech Commands dataset (Warden, 2018) and a two-layer MLP on the tabular Iris dataset (Fisher, 1936). The hyperparameters used for training are defined in the respective sections. All reported results for CIFAR 10/100 are obtained from 15 different seeds and the mean and standard deviation are reported in the respective tables. Due to limited computational resources, the results from ImageNet1K experiments are obtained from 3 different seeds. Thus, we illustrate the mean and standard deviation on the validation performance. Afterward, the table presents the test accuracy of the model with the highest validation performance.

The implementation is based on PyTorch™ v2.0 (Paszke et al., 2019) and as processing unit, we used two NVIDIA® Tesla® A30 GPUs. The implementation is publicly available. [1]

### 5.1 MNIST experiments

In order to demonstrate that a reduction in the quantity of training data results in an increase in the KL divergence, and consequently, a greater deviation from the expected value, an evaluation was conducted on LeNet (Lecun et al., 1998) in conjunction with Benford and L2 regularization across various subsets of MNIST. The Benford regularizer was scaled to a value of 0.1, and the mean test error curves across five seeds are presented in Figure 2. The results indicate that with fewer data points, greater improvements can be achieved through Benford regularization. In addition to the substantial enhancement in test performance when training with only 1% of the data, the efficacy of this regularization diminishes as more data is incorporated into the training process. Concerning the discrepancy from BL, the proposed regularization effectively mitigates the BL KL divergence. However, it does not fully compensate for the absent data. Networks with reduced BL KL demonstrate superior performance. It is noteworthy that the convergence speed of the BL regularized network is accelerated when utilizing only 10% and 1% of the data. However, during training, the BL KL values of both networks become comparable. This observation underscores a discernible correlation between the BL KL and the test error, thereby highlighting the significance of the proposed regularizer in facilitating convergence speed.

---

[1]https://github.com/juliusott/benford_regularizer

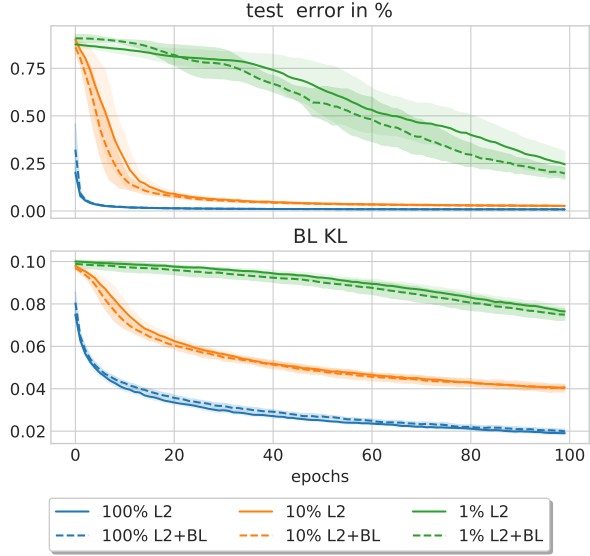

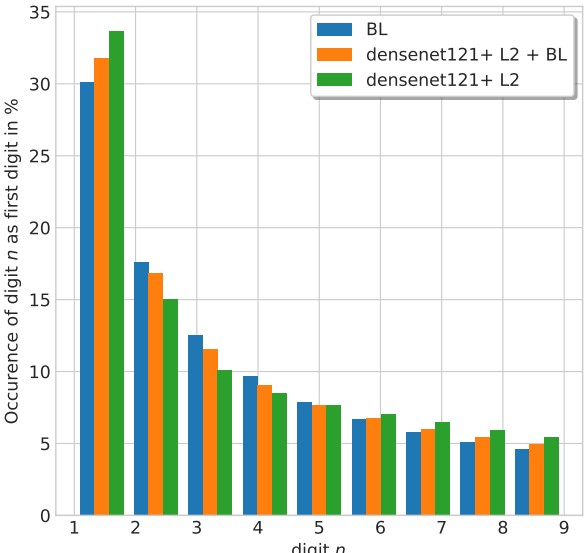

Figure 2: Comparing the test error in % and BL KL of L2 and L2 with Benford regularization when training LeNet on 100%, 10% and 1% of the MNIST dataset.

Figure 3: Comparison between Benford's Law and significant digits of DenseNet121 weights trained with L2 and Benford regularization and with L2 regularization alone on CIFAR 10.

## 5.2 CIFAR experiments

In this study, we initially demonstrate the efficacy of the Benford regularizer on the CIFAR 10 and CIFAR 100 datasets. Both consist of $50,000$ training images and $10,000$ test images with 10 and 100 different classes respectively. A subset of $10,000$ images from the training data is used for validation. The evaluation of different dataset sizes is conducted by maintaining the validation and test data sizes constant. To this end, we present the results of CNN- and Transformer-based models introduced by Vaswani et al. (2017). For this, the DenseNet by Huang et al. (2017) with depths 121 and ResNext with 29 layers by Xie et al. (2017) represent the CNN family. Furthermore, the tiny version of the Swin Transformer by Liu et al. (2021) and the Vision Transformer (ViT) for small datasets proposed by Lee et al. (2021) represent the Transformer models. The input images are normalized to $32 \times 32$ images with a random crop and random horizontal flip during training, and for testing, the images are only normalized. The CNN models are trained with an SGD optimizer, initialized with 0.9 momentum and $5 \cdot 10^{-4}$ weight decay, and an initial learning rate of 0.001. The Transformer models utilize an Adam optimizer with an initial learning rate of 0.0001 and a cosine annealing learning rate schedule. Each network is trained for 200 epochs, and the learning rate of the CNNs is divided by 5 after $[60, 120]$ epochs, following the settings in DeVries & Taylor (2017). For all models, the quantile regression loss for each layer is computed, and the average is added as regularization with a scaling factor of 0.1. As demonstrated in Figure 3, the proposed Benford regularizer not only minimizes the BL KL but also modifies the significant digits of the DNN weights to approach BL. In contrast, a network optimized with L2 regularization only exhibits deviation from BL. The results in Table 1 and Table 2 report the mean and standard deviation of the test error obtained over 15 seeds. The results show that the proposed Benford regularizer improves the network performance when the number of data samples is limited. As an additional illustration, Figure 4 show the improvements of the tiny ViT's and DenseNet121's test performance on the subsets of CIFAR 10 and CIFAR 100 with the additional use of Benford regularization.

It is important to note that the ViT trained with L2 and Benford regularization on 80% of CIFAR 10 achieves a performance level comparable to that of the ViT trained on the entire dataset. Generally, the ViT benefits from the L2 and Benford regularization across all dataset sizes. Given the substantially larger size of Transformers in comparison to the evaluated CNN models, the overall test accuracy is lower. However, the discrepancy between model size and dataset diversity elucidates the ViT's benefits, both on subsets and

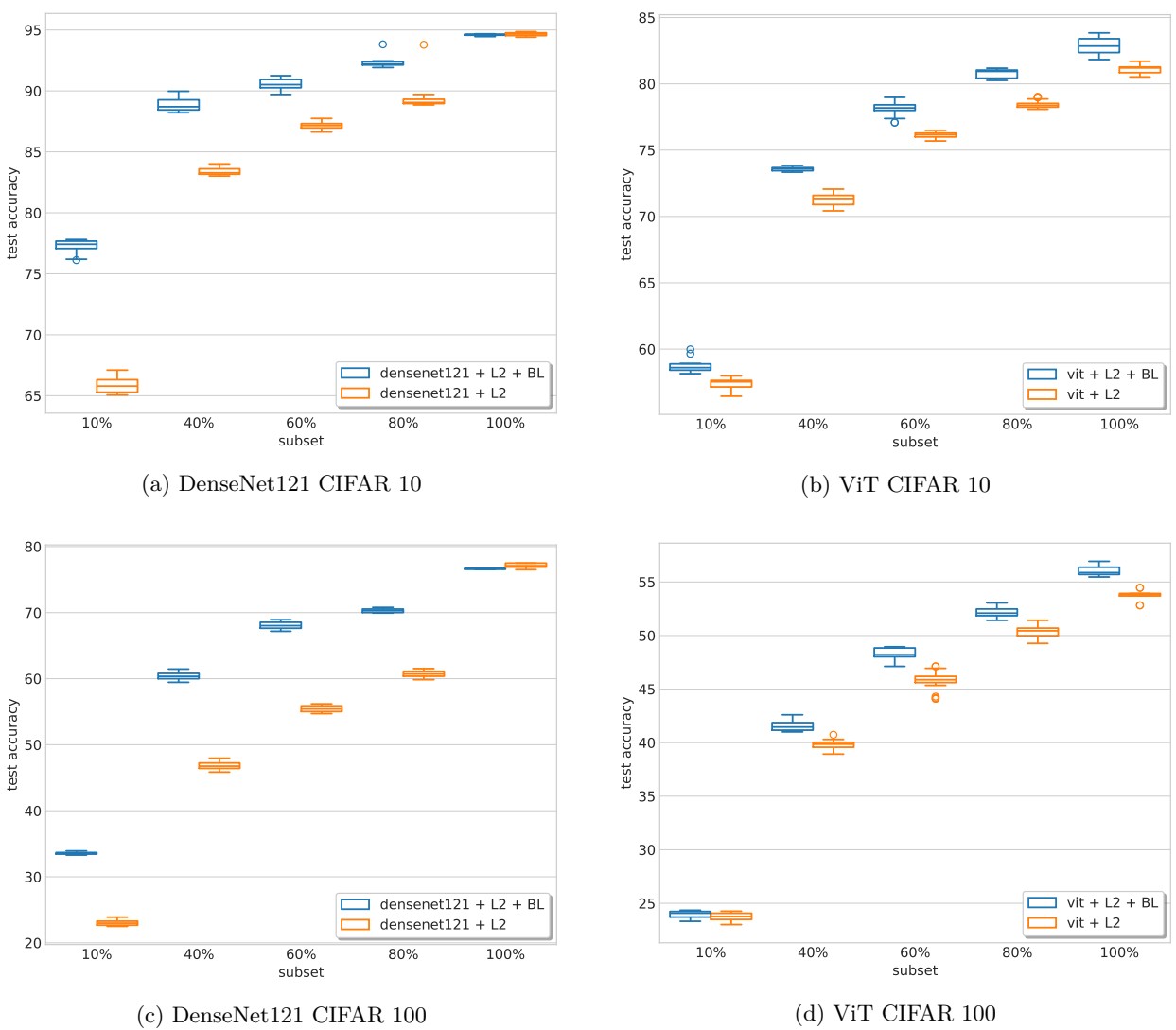

Figure 4: The distribution of DenseNet121's test accuracy in % (↑) trained on subsets of CIFAR 10 (a) and CIFAR 100 (c) shows that it improves with BL regularization on all subsets but achieves the same performance as L2 regularization on the full dataset, respectively. The distribution of tiny ViT's test accuracy on CIFAR 10 (b) and CIFAR 100 (d) improves on the subsets and on the full dataset over L2 regularization.

on the full dataset. Conversely, the effect of Benford regularization becomes negligible when the complexity of the dataset surpasses the capacity of the model and the available data, as evidenced by the performance of ViT on 10% of CIFAR 100.

In contrast, the CNN models exhibit enhancement of up to 15% on the subsets of CIFAR 100 and up to 10% on subsets of CIFAR 10. Additionally, the Benford regularization imposes constraints on the full dataset for the CNN models, where their performance approaches an optimal state for the respective architecture.

## 5.3 Imagenet experiments

In the final set of image classification experiments, the complexity of the task was amplified by employing the ImageNet1K dataset (Russakovsky et al., 2015). The focal point of these experiments lies in the MobileNetV3 networks by Howard et al. (2019), which are optimized for mobile applications. In opposition to the preceding experiments, this section emphasizes reduced models for extensive datasets. To this end, we have trained

Table 1: Average test error in % (↓) and standard deviation on CIFAR 10. An evaluation of the L2 regularization with (L2+BL) and without Benford regularization on the full dataset and random subsets of 80%, 60%, 40%, and 10%. We highlight the respective results in bold when the performance difference is at least one standard deviation.

| | Reg. | Subset | | | | |
|---|---|---|---|---|---|---|
| | | 100% | 80% | 60% | 40% | 10% |
| Swin | L2 | $15.70 \pm 0.35$ | $17.44 \pm 0.48$ | $20.39 \pm 0.46$ | $24.61 \pm 0.55$ | $42.72 \pm 0.72$ |
| Transformer | L2+BL | $15.17 \pm 0.42$ | $16.66 \pm 0.59$ | $19.94 \pm 0.41$ | $24.55 \pm 0.50$ | $41.72 \pm 0.53$ |
| Tiny ViT | L2 | $18.91 \pm 0.35$ | $21.21 \pm 0.35$ | $23.86 \pm 0.22$ | $28.48 \pm 0.41$ | $42.34 \pm 0.59$ |
| | L2+BL | $\mathbf{17.97 \pm 0.40}$ | $\mathbf{19.51 \pm 0.31}$ | $\mathbf{21.66 \pm 0.22}$ | $\mathbf{26.6 \pm 0.36}$ | $\mathbf{41.75 \pm 0.59}$ |
| DenseNet121 | L2 | $5.61 \pm 0.24$ | $10.74 \pm 0.81$ | $12.86 \pm 0.27$ | $16.30 \pm 0.80$ | $33.65 \pm 0.47$ |
| | L2+BL | $5.71 \pm 0.245$ | $\mathbf{7.6 \pm 0.61}$ | $\mathbf{9.01 \pm 0.27}$ | $\mathbf{12.04 \pm 0.74}$ | $\mathbf{27.62 \pm 0.64}$ |
| ResNext29 | L2 | $6.03 \pm 0.19$ | $12.88 \pm 2.85$ | $16.55 \pm 0.35$ | $22.10 \pm 0.55$ | $43.23 \pm 0.39$ |
| 2x64 | L2+BL | $6.03 \pm 0.11$ | $\mathbf{9.04 \pm 0.29}$ | $\mathbf{10.49 \pm 0.38}$ | $\mathbf{13.3 \pm 0.55}$ | $\mathbf{32.35 \pm 0.39}$ |

Table 2: Average test error in % (↓) and standard deviation on CIFAR 100. An evaluation of the L2 regularization with (L2+BL) and without Benford regularization on the full dataset and random subsets using 80%, 60%, 40% and 10% of the training data. We highlight the respective results in bold when the performance difference is at least one standard deviation.

| | Reg. | Subset | | | | |
|---|---|---|---|---|---|---|
| | | 100% | 80% | 60% | 40% | 10% |
| Swin | L2 | $45.96 \pm 0.44$ | $50.33 \pm 1.82$ | $54.94 \pm 1.47$ | $61.11 \pm 1.13$ | $78.08 \pm 0.40$ |
| Transformer | L2+BL | $\mathbf{44.08 \pm 0.47}$ | $\mathbf{48.09 \pm 0.46}$ | $\mathbf{52.41 \pm 0.77}$ | $\mathbf{58.87 \pm 0.70}$ | $77.65 \pm 0.34$ |
| Tiny ViT | L2 | $46.26 \pm 0.53$ | $49.64 \pm 0.52$ | $54.21 \pm 0.81$ | $60.2 \pm 0.40$ | $76.33 \pm 0.43$ |
| | L2+BL | $\mathbf{43.88 \pm 0.36}$ | $\mathbf{48.43 \pm 0.58}$ | $\mathbf{51.64 \pm 0.80}$ | $\mathbf{57.89 \pm 0.27}$ | $76.28 \pm 0.47$ |
| DenseNet121 | L2 | $22.71 \pm 0.70$ | $39.25 \pm 0.47$ | $44.55 \pm 0.48$ | $53.18 \pm 0.49$ | $77.09 \pm 0.60$ |
| | L2+BL | $23.36 \pm 0.08$ | $\mathbf{29.7 \pm 0.23}$ | $\mathbf{32.88 \pm 0.46}$ | $\mathbf{39.51 \pm 0.64}$ | $\mathbf{66.43 \pm 0.72}$ |
| ResNext29 | L2 | $22.97 \pm 0.14$ | $43.02 \pm 0.33$ | $49.81 \pm 0.39$ | $59.11 \pm 0.44$ | $80.38 \pm 0.31$ |
| 2x64 | L2+BL | $\mathbf{22.62 \pm 0.13}$ | $\mathbf{29.99 \pm 0.35}$ | $\mathbf{35.28 \pm 0.9}$ | $\mathbf{42.61 \pm 0.46}$ | $\mathbf{74.5 \pm 0.37}$ |

the small and large versions of MobileNetV3 on the Imagenet1K dataset with default data augmentation techniques, including horizontal flipping, random cropping, and color jittering. We have scaled the Benford regularization loss with a factor of $10^{-6}$, which has yielded optimal results. The networks were trained for 90 epochs, which was the convergence limit, with an initial learning rate of 0.1, which was divided by 10 after 30 and 60 epochs. Due to limited resources, networks were trained with a batch size of 256 images, while networks proposed in the original work were trained with a batch size of 4096. As illustrated in Figure 7, the mean and standard deviation of the validation error were calculated over three runs, with the best validation performance of each network used for testing. As shown in Table 3 and the validation error curve, the proposed regularization method resulted in a significant improvement.

Table 3: Imagenet1K test accuracy in % (↑) for the MobileNetV3 networks.

| | Regularization | |
|---|---|---|
| MobileNetV3 Large | L2 | 66.96 |
| | L2+BL | **67.82** |
| MobileNetV3 Small | L2 | 59.3 |
| | L2+BL | **59.77** |

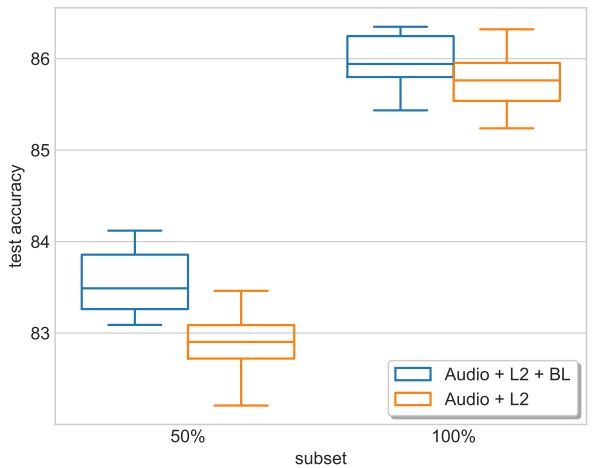

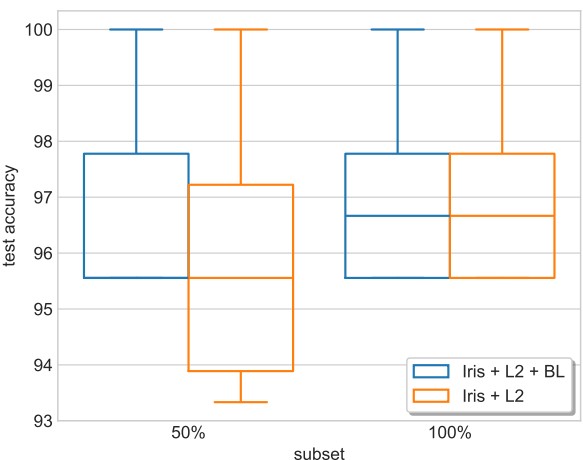

Figure 5: Distribution of the test accuracy in % (↑) trained on 100% and 50% subsets of Google Speech Commands (Audio) dataset.

Figure 6: Distribution of the test accuracy in % (↑) trained on 100% and 50% subsets of the tabular IRIS dataset.

## 5.4 Speech and tabular datasets

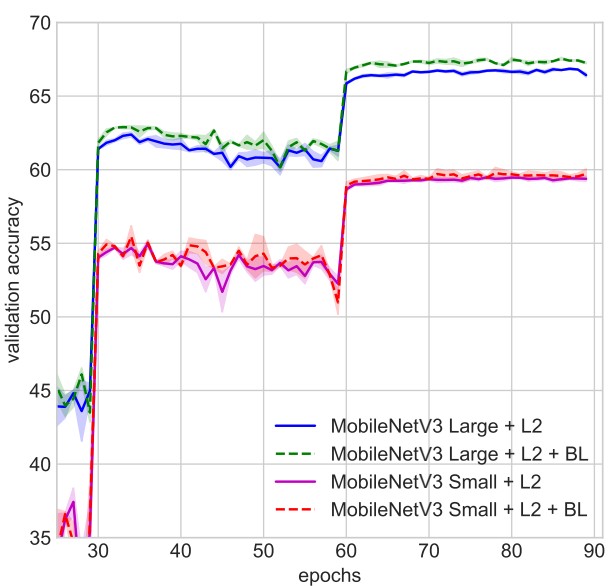

Figure 7: Validation accuracy (↑) of MobileNetV3 small and large. Both models benefit from the Benford regularization.

To further demonstrate the effectiveness of the Benford regularizer, we propose the evaluation of different data domains. For this purpose, we train the M5 model of Dai et al. (2017) on the Google Speech Commands dataset (Warden, 2018). The datasets contain recordings for a total of 85,511 training, 10,102 validation, and 4,890 test recordings. For training, we use the Adam optimizer with a weight decay of 0.01 and an initial learning rate of 0.1 and scale the Benford regularizer by a factor of 0.1. The model is trained for 100 epochs, and the learning rate is divided by 10 after 20 epochs, as described in Dai et al. (2017). For the experiments on the IRIS dataset (Fisher, 1936), we use a neural network consisting of two fully connected layers with 10 and 3 neurons and a ReLu activation function (MLP). The dataset consists of 150 samples, randomly divided into 73 for training, 32 for validation, and 45 for testing. The model is trained for 1000 epochs with the Adam optimizer and an initial learning rate of 0.001. The Benford regularization is scaled by a factor of 0.001. The results in Table 4 show that the Benford regularization also improves the performance of DNNs regardless of the data domain. Looking at Figure 5, in the case of the more complex audio dataset, the Benford regularization improves the average performance of the model regardless of the dataset size. The enhancement in performance achieved through training on a mere 50% of the audio data is substantial, with an average improvement of 10% and a reduced variability. For the more straightforward IRIS dataset, Figure 6 presents a boxplot illustrating that the performance of the 50% subset is enhanced by Benford and L2 regularization, while the performance of the full dataset remains nearly unchanged. In this scenario, the network approaches an optimal state.

## 6 Limitations

The experimental findings indicate that regularizing the DNN weights towards BL improves the overall performance, particularly in scenarios with limited training data. The proposed regularization method consistently enhances performance in conjunction with L2 regularization. As illustrated in Figures 2 and 3, the network weights can be optimized to approach BL. Given the scale invariance of Benford regularization, there are infinitely many ways to approximate it, which has led to the use of L2 and Benford regularization in tandem. Furthermore, experimental findings have demonstrated that Benford regularization is constrained by the amount of data, the complexity of the data, and the model capacity. In instances where the model is unable to learn reasonable features, Benford regularization is ineffective in compensating for this deficiency. Additionally, when the network is presented with sufficient data and it is near optimum, the Benford regularization has limited effect on the distribution of the weights because it is already closely following the Benford distribution. Since the applied error bounds are related to exponential functions, such as the softmax function, the experiments were carried out on classification tasks. Energy-based models learn an exponential energy function, which is used for generative modeling. This finding suggests that our approach is transferable to generative modeling, but not to regression problems, necessitating further analysis in this area.

Table 4: Average test error ($\downarrow$) and standard deviation on tabular and speech datasets. An evaluation of the Benford regularization on the full datasets and a random subset of 50%.

|  | Regularization | Subset | |
| --- | --- | --- | --- |
|  |  | 100% | 50% |
| M5 (Audio) | L2 | $14.12 \pm 0.39$ | $17.14 \pm 0.38$ |
|  | Benford | $14.11 \pm 0.31$ | $\mathbf{16.04 \pm 0.35}$ |
| MLP (IRIS) | L2 | $3.43 \pm 0.049$ | $5.5 \pm 0.05$ |
|  | Benford | $\mathbf{3.12 \pm 0.046}$ | $\mathbf{4.22 \pm 0.03}$ |

## 7 Conclusion and future work

While previous approaches only observed whether obtained data follows BL, this paper is the first to propose the Benford regularization, where BL is learned via gradient-based optimization. The motivation is based on commonalities between DNNs, BL and thermodynamics and the intriguing features of BL. It is scale invariant, thus independent of the data domain and is commonly used to detect bias and anomalies in measured datasets. The proposed method applies Benford regularization in combination with L2 regularization for numerical stability and presents substantial improvements. Extensive experiments were conducted on random subsets of common image datasets CIFAR 10/100 with CNN and Transformer-based models of varying number of parameters. The addition of Benford regularization boosted the performance up to 15% on subsets of CIFAR 10 and up to 10% improvement on CIFAR 100 subsets. The limitations are observed when the model either reaches the optimum or the dataset is too complex for the limited amount of data. In these cases, the performance is the same as training with L2 regularization alone. Consequently, experiments on low-capacity MobileNetV3 models are evaluated on the large and complex ImageNet1k dataset. As expected, the larger capacity network benefits more from the additional Benford regularization as the smaller version due to the larger amount of available data. This observation is further shown by additional experiments on the tabular Iris and Google Speech commands datasets.

To summarize, our experiments demonstrate an interplay between model and dataset complexity, revealing that Benford regularization yields the most significant improvements when a large-capacity model is paired with limited data or a lower-capacity model is paired with abundant data. However, we observe that there are limits to this approach, as overly complex models for simple data (e.g., Swin Transformer) or low-capacity models for complex data (e.g., MobileNetV3 Small) cannot effectively leverage this regularization.

These results not only provide insights about model capacity but have practical implications for industrial DNN applications where data collection is challenging and model sizes are constrained by hardware design.

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

# A    Appendix

## A.1    Ablation on weight decay parameters

In this set of experiments, the effect of weight decay magnitude on performance is studied. Due to the small gap between the performance of L2 and regularized tiny ViT, this model is chosen for the ablation study. The objective is to determine whether a stronger L2 regularization can close the gap to the regularized version using BL. The CIFAR 100 experiments in Table 2 utilize a weight decay rate of $5 \cdot 10^{-4}$. Consequently, the experiments in Table 5 employ $10^{-4}$, $10^{-3}$, and $10^{-2}$ to assess the impact of varying weight decay magnitudes. In general, the tiny ViT benefits from larger weight decay parameters, a phenomenon that is also observed in the regularized version of the BL. In the final experiment, the L2 regularization was removed. As anticipated, the elimination of L2 regularization led to a decline in the performance of the regularized version of BL, given BL's inherent scale invariance, which necessitates its use in conjunction with L2 regularization. In summary, the efficacy of weight decay in enhancing performance when confronted with constrained datasets is noteworthy. However, it does not surpass the regularized version of BL when employing equivalent training parameters.

Table 5: Average test error ($\downarrow$) and standard deviation on CIFAR 100 for different weight decay magnitudes.

|  | decay rate | Subset | | |
| --- | --- | --- | --- | --- |
|  |  | 80% | 40% | 10% |
| Tiny ViT | $10^{-4}$ | $49.34 \pm 0.34$ | $60.09 \pm 0.58$ | $76.23 \pm 0.16$ |
|  | $10^{-3}$ | $49.58 \pm 0.73$ | $59.69 \pm 0.016$ | $76.14 \pm 0.27$ |
|  | $10^{-2}$ | $48.76 \pm 0.41$ | $59.16 \pm 0.47$ | $75.83 \pm 0.30$ |
| Tiny ViT + BL | $10^{-4}$ | $47.84 \pm 0.41$ | $58.07 \pm 0.26$ | $76.37 \pm 0.12$ |
|  | $10^{-3}$ | $47.48 \pm 0.46$ | $58.18 \pm 0.39$ | $75.44 \pm 0.37$ |
|  | $10^{-2}$ | $47.02 \pm 0.68$ | $58.17 \pm 0.69$ | $75.39 \pm 0.22$ |
| Tiny ViT + BL only | 0 | $49.84 \pm 0.41$ | $59.82 \pm 0.43$ | $76.47 \pm 0.14$ |

