# OpenReview forum: "Exploiting Benford's Law for Weight Regularization of Deep Neural Networks"
_TMLR — Accepted by TMLR_

### Review · Reviewer_3i5Z · 2024-12-13

**Summary Of Contributions:**

this paper investigates the use of Benford's Law as a technique to conduct weight regularization in DNNs. The authors' approach is based on a recent finding by Sahu et. al that significant digits can be seen as a predictor of the generalization of DNNs to the validation data and can be used as early stopping criterion. Gradient-based optimization is practically used to conduct the regularization of weights.

**Audience:**

Yes

**Claims And Evidence:**

No

**Requested Changes:**

* make contribution clearer. for me it reads like a refinement of L2 regularization that can help in specific architecture choices in classification tasks. However, the lack of theory and missing benchmarks does not support the currently made general claim. So I suggest to either extend the experiments or to narrow down the contribution to what the authors can show empirically.
* language needs to be improved and mathematical inconsistencies correct

**Strengths And Weaknesses:**

# Strengths

* Relevant topic: training and regularizing DNNs with limited data
* experiments to verify claims


# Weaknesses

## approximation error
* why is the fact that the union of multiple datasets follows BL is sufficient and how is it generalizable? one can always find a set of data sets where neither the distinct ones nor their union follows BL closely?


## experiments
* means and standard errors are based on 3 different seeds only. I consider this too little to obtain robust results
* the only competitors are L2 regularization and L2+BL which does not seem convincing to me without benchmarking against other methods
* when using transformer architectures the improvements seems rather marginal which indicates that the proposed refinement works only for specific architectures
* similar in section 5.4 the improvement of BL is not really present in particular for small data sets


## further comments
* is (9) related to isotonic regression and post-calibration approaches?
* am I correct that the proposed approach only works when being used on-top of the usual L2 regularization? why L2 regularization and not other weight regularization methods for DNNs?

## mathematical flaws
* equation (3) appears wrong. it should be either BL on the right-hand side or assumed that P is a distribution that follows BL exactly.
* please increase brackets in (1) and (2)
* the rounding operator in (6) is not properly introduced
* the arguments of the uniform distribution below (8) should be non-bold


## Language
* the paper contains a number of grammatical errors and typos, e.g. "gradient base optimization" or " significant digits is "
* some sentences are hard to follow and language should be improved.

---

> ### Author Response · Authors · 2024-12-18
> **Response to Reviewer 3i5Z**
>
> Dear Reviewer, \
> Thank you for your valuable feedback and suggestions regarding our manuscript. In the following,
> we would like to respond to your feedback point by point.
>
> ### **Approximation error:**
> The similarity between Benford's Law (BL) and the significant digit distribution from the union of
> many datasets has been observed in many cases and serves as a motivation for our regularizer. In
> fact, it is the case that some distributions, such as those described in "Benford's law hits back: No
> Simple Explanation in Sight for Mathematical Gem" (Berger and Hill, 2011), do not follow BL.
>
> ### **Experiments:**
> We welcome your feedback on our experiments. There seems to be some confusion about the comparison to the L2 regularized networks. We would like to clarify that the L2 regularized networks refer to the default training mechanism for each network. This includes the use of weight decay as L2 regularization, standard data augmentation, and layer normalization. \
> For a fair comparison, we looked at weight regularization using penalty terms that work directly on the weight distribution. These include L2, L1, or total variation regularization. Other regularizers, such as layer normalization or data augmentation, affect the weight distribution indirectly. \
> Since L2 regularization is used in all standard training settings, we compare the Benford regularizer to the best performing regularizer with a penalty term. Our approach adds a penalty term to the standard training procedure and evaluates whether it improves the performance of the networks. It is reasonable to use our loss in addition to the L2 penalty term because it does not take numerical stability into account. To further support the significance of our experiments, our CIFAR 10/100, speech, and table experiments consider 15 different seeds. Only the ImageNet experiments were performed with 3 seeds due to limited computational resources. \
> The exact improvements in absolute terms depend on the network architecture. The convolutional networks benefit from the inductive bias, while the transformers do not. We also mention in our discussion that the complexity of the task plays an important role in the regularization effect. In particular, some networks fail to learn useful features, and no regularization term is able to compensate for this. However, our Benford regularizer consistently improves performance regardless of the dataset.
>
> ### **Further comments:**
> We thank the reviewer for pointing this out. Since the weights are ordered before the regression
> loss, Equation 9 is related to an isotonic regression with equal weights for all data points.
> Mathematical flaws and language:
> We thank the reviewer for pointing out that the readability can be improved. We want to clarify
> that P in equation 3 is exactly the distribution of BL described in equation 1. We will also update
> the remaining equations based on your suggestions. Finally, the manuscript will be revised to
> improve readability and remove spelling errors.
>
> ### **Requested Changes:**
> We appreciate your feedback on the potential misunderstanding between the experiments and
> the claims of the paper. We clarify that our approach adds a penalty term in addition to the
> standard L2 regularization, which is particularly useful for training with limited data. In addition,
> we specify that the improvements depend on the network design and task complexity.
> As mentioned in the previous point, the language of the manuscript is revised to improve
> readability. \
> Sincerely, \
> The Authors

---

### Review · Reviewer_Epf9 · 2024-12-17

**Summary Of Contributions:**

The paper claims that the deviation of the significant digit distribution of DNN weights from Benford’s Law (BL) would affect the generalization of trained models. Based on some previous works, the paper introduces BL regularization to the training of neural networks. Experiments on images, speech, tabular data etc. are presented to demonstrate that adding the regularizer helps improve model accuracy.

**Audience:**

Yes

**Claims And Evidence:**

Yes

**Requested Changes:**

1. Some sentences may have typos. I highlight the correction of potential typos in boldface.

    i) “this bias **is** reflected in the weights of DNNs”.

    ii) “where $**f(x,\theta)**$ defines the neural network output”.

    iii) “return **bl_loss**”.

2. In eq (2), what is the $X$ mentioned in “$log⁡_{10} (X)  mod 1 \sim U(0,1)$"?

3. The description of the algorithm should be clearer. For example, I am confused about

    i)  What is the MSE loss mentioned in the pseudo algorithm (is it just eq (9)? But eq (9) is not MSE)?

    ii)	How is the comparison between quantile and uniform distribution related to the distribution of significant digit?

    iii) Why mention the error function from Engel & Leuenberger (2003)? How is it related to this paper (especially considering the fact that Engel & Leuenberger considered exponential random variables but not having a neural network setting). Why we use exponential random variable here? And in the definition eqs. (10), where does the temperature τ come from (is there a specific application behind it)?

    iv) In the algorithm, what does $N$ refer to? And is there any word after the first "#"?

Thus, I strongly suggest the authors reorganize this description of the algorithm and make clear about the notations they use. For example, it would be much clearer if the complete formula for the loss function is presented.

4. The authors did a wide range of experiments to demonstrate that adding BL regularizations would improve the accuracy of models. I suggest the authors further discuss when to add the BL regularizer, considering different models and different data size. For example, (a) on CIFAR 10/ CIFAR 100, the improvement of accuracy is small for a large portion of dataset, could you please explain why the performance differs on different models? (b) The experiments on Imagenet, speech and tabular data show only about 1% improvement, so is it still necessary to use BL regularization?

5. As the authors mentioned in Section **6 Limitations**, the experiments consider errors based on Softmax function. It would be great if the authors give some explanations/guess on how the structure of training and testing loss functions affect the BL KL, or why it does not.

**Strengths And Weaknesses:**

Strength: The idea of considering BL for weights is novel and interesting to me. Moreover, a large number of experiments are presented to support the algorithm proposed by the authors.

Weakness: There could be more reasoning, and explanations for the algorithm; it may also be helpful to discuss how we may apply it in practice. Besides, some parts of the paper are not written in a clear way. See my comments below.

---

> ### Author Response · Authors · 2024-12-22
> **Response to Reviewer Epf9**
>
> Dear Reviewer, \
> Thank you for the valuable feedback and suggestions to improve our manuscript. We appreciate the acknowledged novelty and the extensive experiments. In the following, we would like to address your suggestions point by point.
> ### **Response to Point 1:**
> First of all, we thank the reviewer for pointing out the typographical errors, which we will correct in the revised manuscript.
> ### **Response to Point 2:**
> We agree that the description of Equation 2 lacks an explanation of the variable X. In the revised manuscript, we add that in Equation 2, X refers to a set of numbers. This can be the pixel values or the FFT coefficients. For the regularizer, X refers to the weights of the neural network. Thank you for pointing out this missing explanation.
> ### **Response to Point 3:**
> Thank you for pointing out the inconsistency between the algorithm and equation 9. In the equation, we present quantile regression based on a distance metric. In the implementation, we use the squared distance. We will update Equation 9 so that it is consistent with the algorithm and the implementation. \
> Furthermore, it has been proven that Benford's Law (BL) is satisfied if $\log_{10}(X)\ mod \ 1 $ is uniformly distributed between $[0,1)$, where X are the weights of the neural network. Details are given in “Benford's Law: Theory and Applications.” by Miller, S. (2015).  \
> Thus, we can apply the logarithm and modulo operator to the neural network weights, which are now also distributed between $[0,1]$. In the text step, we want to compute the distance between the weights and the uniform distribution, which is done in Equation 9. \
> In "A concise proof of Benford's law" (Wang & Ma, 2023) the authors observe an oscillation around BL. A similar observation was made in "Rethinking Neural Networks with Benford's Law" (Sahu et al., 2021).  \
> This suggests that a perfect match between the significant digits of the weights and BL is not desirable, but should be within error bounds. Those proposed by Leuenberger & Engel are of particular interest because they consider two facts related to neural networks. First, they consider exponential functions of the form $f(x) = \lambda * e^{-\lambda x}$ and the error bound is independent of x and lambda. Thus, these bounds can be applied to a classification neural network without further adjustment since it is an exponential function. The temperature parameter was introduced to bridge the general exponential function to that used in NNs, and N is the number of weights used in the loss computation. \
> We agree that the wording regarding the exponential variable is misleading in this case and will revise the manuscript based on the response given.
> ### **Response to Point 4:**
> For the careful evaluation of our experiments we would like to express our gratitude. We consider convolutional and transformer neural networks. Convolutions are known to process less data more efficiently due to their inductive bias. This seems to be related to the effectiveness of our regularizer. On the contrary, transformers are less efficient with respect to the data and suffer more from reduced datasets. However, the improvements are very consistent. The actual improvements depend on factors such as dataset size, model architecture, number of parameters, and dataset complexity. Regarding the performance gain on ImageNet or the Tabular Iris dataset, it is important to consider the standard deviation, which is very small. This indicates that the improvements are relevant and due to our regularizer. This discussion is added to the experimental section for a better reader experience.
> ### **Response to Point 5:**
> Thank you for the detailed questions about our algorithmic choices. We would like to emphasize that this work considers neural networks that represent an exponential function, such as softmax. \
> Alternatively, they can be formulated as an energy-based model that represents an exponential energy function. Energy-based models are also used for image generation. The close relationship between BL and the exponential family suggests that other training losses also have solutions close to BL. \
> Another interesting question is whether optimization processes generally produce solutions close to BL. This experimental and theoretical evaluation remains to be done in future work. In addition, we add the relation to energy-based models in the updated manuscript. \
> We appreciate your profound questions and comments, which help us to improve our work.
>
> Sincerely, \
> The Authors

---

### Review · Reviewer_UuAS · 2024-12-26

**Summary Of Contributions:**

This paper introduces a novel perspective on neural network regularization with cross-domain, cross-architecture applicability. The work is well-structured and follows a clear logical flow from theoretical motivation to practical implementation. While the approach shows most benefit in specific scenarios (limited data, matching model capacity), this specialization doesn't make it less valuable as I think many real-world applications face exactly these constraints. The technical contribution of making Benford's Law suitable to gradient-based optimization is significant.
Main contributions are:
- discussing Benford's Law connection to neural network weights (but this is also done in other works, as the authors have mentioned in the menuscript)
- devising a method to use the Benford's prior to regularize the model, with experiments on different domains.

**Audience:**

Yes

**Broader Impact Concerns:**

No concerns.

**Claims And Evidence:**

Yes

**Requested Changes:**

**Questions/Doubts**:
- Only L2 and L2+BL results are shown, but BL-only baseline is missing, why?
- What is the computational cost or memory overhead compared to standard regularization methods (e.g. L2)

**Suggestions**:
- The "Approach" may be rewritten with more intuitive explanations and can benefit from some polishing.

**Strengths And Weaknesses:**

**Strengths**:
- The paper tackles regularization, a fundamental problem relevant to all neural networks.
- The regularization method is novel, being the first to use Benford's Law for weight regularization rather than just observation.
- The approach is well motivated through connections to other disciplines. The motivations section is really enjoyable to read because it builds from statistical principles to neural network applications.
- Results are comprehensive and convincing. The validation is done across multiple domains (image, speech, tabular) and testing on diverse architectures.

**Weaknesses**:
- Limited benefits in optimal conditions, like when models are already well-performing, or when sufficient training data is available. In these cases, performs similarly to standard L2 regularization
- Current theoretical framework mainly applies to classification tasks
- The "Approach" section could benefit from more intuitive explanations before the mathematical formulations:
  - The introduction of the fractional part (eq. 6) needs better connection to eq. 2
  - The quantile regression method could use more intuitive motivation (not sure how feasible this is though)

---

> ### Author Response · Authors · 2024-12-30
> **Response to Reviewer UuAS**
>
> Dear Reviewer, \
> Thank you very much for the positive feedback on our work and the highlighted novelty. To further improve the readability, the "Approach" will be polished with more intuitive explanations in the revised manuscript. \
> The following is a point-by-point response to your concerns and questions.
>
> ### **Response Point 1:**
> Thank you for your careful review of our experiments. The limited improvements on the full datasets are to be expected, as the bias is reduced by more data. Our approach targets biased combinations of dataset size and limited model capacity. We include these points in our limitations section and the discussion of the experiments.
>
> ### **Response Point 2:**
> We appreciate your thorough evaluation of the theoretical basis. We agree that the proposed concept currently applies to classification tasks, since we model exponential functions there. However, it is noteworthy that generative models also model exponential functions, which could be a bridge to other applications. These will be considered in future work.
>
> ### **Response Point 3:**
> We are very grateful for the suggestions to improve our manuscript and agree that the connection between Eq. 2 and Eq. 6 is missing. We will add that the module 1 operator is identical to the fractional part of a number.
>
> ### **Response Question 1:**
> Thanks for the deep question. There are two reasons why a BL-only regularization would fail. First, BL is scale independent. So we could make any network arbitrarily close to BL, but it would not give reasonable results due to lack of numerical stability. In addition, the default training scheme considers L2 regularization and this is what we want to use as a baseline.
>
> ### **Response Question 2:**
> Thanks for the question to improve the clarity. In the approach we mention that the average complexity is O(n). The quantile computation is based on the Quickselect algorithm, which has an average time complexity of O(n). We will add the origin of the time constant in the revised manuscript.
>
> We thank the reviewer for the suggestions to improve the quality of the approach and the questions to clarify our algorithmic decisions. We will revise the manuscript accordingly.
>
> Sincerely, \
> The Authors

---

### Comment · Action_Editor_LXyM · 2025-01-06

Dear all,

A brief reminder that the rebuttal period will close soon. I invite all reviewers to acknowledge the authors' answers and let them know if you have lingering questions or comments that can be solved / discussed in a short timeframe, before the final decision.

Thanks,
The AE

---

### Decision · Action_Editor_LXyM · 2025-01-24

**Recommendation:** Accept with minor revision

**Comment:**

We received three reviews on the submission. Reviewers provided a point-by-point response to most comments, although the reviewers did not interact further during the rebuttal period. At the end, 2 out of 3 reviewers are favourable, while 1 reviewer leans towards a rejection. There is a consensus that the technique presented in the paper is novel and might be of interest, as it touches a core topic in neural networks. The third reviewer, however, is concerned that the proposed experiments are limited, as they only consider a single baseline (l2 regularization).

Overall, I believe the paper is suitable for acceptance, but I would ask the authors for a few improvements in the camera ready version, based on the reviewers' comments. I believe these improvements are necessary for a complete validation of the algorithm, especially points 3 and 4.

1.  The language quality must be significantly improved, as there are many errors (e.g., "making them prune to overfit"), or sentences that do not make sense in English ("To the best of the authors notice"). Please proofread the manuscript professionally.
2. Math should be improved, especially by avoiding the inconsistent use of terms such as $log$ and $\text{log}$ (I suggest strongly to stick to the latter).
3. One reviewer was concerned about the lack of experiments using only BL. While the answer seem convincing, I believe adding a small ablation in this sense (with an explanation of the results), possibly as an appendix, could be beneficial.
4. The experiments consider a single setup for the weight decay and BL weights. I believe it is necessary to add an experiment showing the change in performance as these parameters are varied (e.g., can we recover the gap with BL by simply increasing the weight decay coefficient?).

**Audience:**

The paper has a wide audience, as it targets a fundamental topic in neural networks (regularized training). While I do not think that the current set of experiments is of immediate interest for the community (as the setups are small, and the results sometimes marginal), the approach is novel and might lead to further interesting developments.

**Claims And Evidence:**

The paper proposes a novel weight regularization technique for neural networks based on the so-called "Benford law" (BL), an empirical law that describes the distribution of the most significant (leftmost) fractional digit in a set of numbers. It extends previous results on the link between BL and generalization by providing a differentiable regularizer that can be included during the training process, which involves an MSE loss over the quantiles of the weights. Experiments are done on a wide set of benchmarks (classification of images, tabular datasets, speech), and different architectures (CNNs, transformers), but only comparing l2 regularization with a mixture of l2 and the proposed BL regularization. Results show improvement (of variable impact), which become significant as the training data is reduced. These results are enough to validate the proposed technique (if the additions proposed below are incorporated).